# A Novel Adaptive PID Controller Design for a PEM Fuel Cell Using Stochastic Gradient Descent with Momentum Enhanced by Whale Optimizer

Mohammed Yousri Silaa [1,*] , Oscar Barambones [1,*] and Aissa Bencherif [2]

1   Engineering School of Vitoria, University of the Basque Country UPV/EHU, Nieves Cano 12, 1006 Vitoria, Spain
2   Telecommunications Signals and Systems Laboratory (TSS), Amar Telidji University of Laghouat, BP 37G, Laghouat 03000, Algeria
*   Correspondence: silaa.mohammed.yousri@gmail.com (M.Y.S.); oscar.barambones@ehu.es (O.B.)

**Abstract:** This paper presents an adaptive PID using stochastic gradient descent with momentum (SGDM) for a proton exchange membrane fuel cell (PEMFC) power system. PEMFC is a nonlinear system that encounters external disturbances such as inlet gas pressures and temperature variations, for which an adaptive control law should be designed. The SGDM algorithm is employed to minimize the cost function and adapt the PID parameters according to the perturbation changes. The whale optimization algorithm (WOA) was chosen to enhance the adaptive rates in the offline mode. The proposed controller is compared with PID stochastic gradient descent (PIDSGD) and PID Ziegler Nichols tuning (PID-ZN). The control strategies' robustnesses are tested under a variety of temperatures and loads. Unlike the PIDSGD and PID-ZN controllers, the PIDSGDM controller can attain the required control performance, such as fast convergence and high robustness. Simulation results using Matlab/Simulink have been studied and illustrate the effectiveness of the proposed controller.

**Keywords:** stochastic gradient descent with momentum; stochastic gradient descent; PID controller; whale optimization algorithm; proton exchange membrane fuel cell

## 1. Introduction

### 1.1. Motivations

The climate has changed throughout history; most of these slight changes have been caused by a small variation in the earth orbit characterized by an abrupt increase in the temperature [1]. Various effects have already begun to appear in various parts of the world, causing serious problems in the natural environment and peoples' lives. In order to remedy this matter and reduce its risks, all over the world, coal plants are being replaced by solar panels, solar thermal energy, wind turbines, and hydrogen power sources [2]. This latter option may be the key to storing renewable energy and could, therefore, change the face of the energy transition. Hydrogen accounts for 75% of all matter and is thought to be one of the three elements that were created in the big bang; it accounts for over 90% of all atoms in the universe and on earth, and is easy to produce from many compounds, for instance, water [3]. The cost of producing hydrogen from renewable energy is anticipated to slump by 50% by the middle of this century, and this could clear the way for even more green hydrogen [4]. Furthermore, people may one day be able to produce their own hydrogen at home [5]. The possibility has encouraged scientists and researchers to look towards to hydrogen cells. Proton exchange membrane fuel cells (PEMFCs) are among the most efficient electrical generators due to their properties such as high energy density, high performance, and high robustness, resulting in multiple applications such as in mobiles, cars, aircraft, and space shuttles [6–10]. The development of PEMFC systems for multiple applications has grown in recent years, and researchers have suggested a diversity of control

techniques from different strategies in order to improve the net output power. However, to extract the maximum net power, DC/DC converters are required to improve the efficiency of the PEMFC stack and to minimize the decrease in power generation efficiency due to the fluctuation of temperature and gas pressures [11].

### 1.2. State of the Art

In order to obtain greater power conversion from cells, many control techniques and algorithms have been adopted in the literature, such as neural network control (NNC) [12], sliding mode control (SMC) [13], fuzzy logic control (FLC) [14], adaptive control [15], maximum power point tracking (MPPT) techniques [16], etc. Hence, in Ref. [17], a conventional SMC was used in a comparison of classical proportional–integral (PI) linear controllers. The proposed technique showed good results in terms of robustness against the extreme load variation. However, despite these results, the SMC still experiences the chattering phenomenon. Hu Peng et al. [18] established a dynamic model of the temperature mechanism of PEMFC based on control and designed a two-dimensional incremental fuzzy controller for the temperature of PEMFC with an integral link according to the established temperature model and empirical control rules. The results show that the model can simulate the dynamic characteristics of PEMFC, and when the temperature of PEMFC is controlled within the ideal working range, the designed controller can be used to control the temperature of PEMFC in real-time; it also has strong robustness. In Ref. [19], a fractional-order proportional–integral-derivative (FOPID) controller was applied to a four-switch buck-step-up DC/DC converter in order to stabilize the PEMFC output power. Simulation results showed that the proposed method achieved better performance than the integer-order controller. Silaa et al. [20] used a high-order sliding mode to keep the PEMFC system working at an efficient power point and as a solution for the chattering phenomenon. Experimental results showed that the proposed control provides a satisfactory result in a terms of reducing the chattering effect by up to 84%. However, despite these results, the proposed controller has a high-power overshoot against the fluctuating load variation. In Ref. [21], an MPPT using particle swarm optimization (PSO) combined with a proportional integral derivative (PID) controller was compared with the perturbing and observing (P&O) technique and sliding mode controller for a PEMFC. The simulation results showed that the proposed MPPT technique achieves a low overshoot, short response time, and low oscillations around the MPP. Ahmed et al. [22] designed a hybrid system consisting of wind and photovoltaic systems as the main source of energy. The fuel cell is installed as a piece of secondary equipment to guarantee a continuous power supply and to address the erratic nature of wind/photovoltaic supply. Derbeli et al. [23] used a high-order sliding mode compared to a conventional SMC to keep the PEMFC operating at a reference current. Results showed that the proposed algorithm was able to reduce the chattering impact by more than 82%, along with providing robustness against the load variation. In Ref. [24], an adequate power point was obtained using a PID controller. The PID controller was programmed to power the fuel cell by changing the boost converter's pulse-width modulation (PWM). Results showed that the goal was accomplished by the proposed approach with better dynamics and good tracking efficiency. In Ref. [25], a PSO based on fuzzy logic (FL) was planned to keep the PEMFC running at an optimal power point. Simulation results showed a power overshoot and undershoot of more than 63%. Fan Liping et al. [26] established a mathematical model of a PEMFC and designed an adaptive FL controller for constant power fuel cells. Experiments show that the designed controller can attain a constant power output of from the PEMFC. Li et al. [27] proposed a fuzzy sliding mode controller to control the air supply flow to the PEMFC stack. The proposed controller has good robustness. The fuzzy sliding mode controller embeds a fuzzy logic inference mechanism in the conventional SMC, resulting in smooth control. The results show that the fuzzy synovial controller eliminates the chattering phenomenon of the traditional synovial controller. The comparison proves that the fuzzy synovial controller can significantly improve the control performance of a PEMFC. In Ref. [28], a PID optimized

by the grey wolf optimizer (PID-GWO), FOPID optimized by the grey wolf optimizer (FOPID-GWO), and a PID optimized by an extended grey wolf optimizer (PID-EGWO) were used to control a DC/DC boost converter linked to a PEMFC. Simulation results show better dynamics, good tracking efficiency, and faster convergence to the optimal solution. In Ref. [29], a model predictive control (MPC) method was designed for a DC/DC boost converter to keep a PEMFC working at an efficient power stage. Experimental results showed that the MPC technique is superior to the PI technique in terms of tracking accuracy, overshoot, and undershoot.

### 1.3. Contributions

The main contribution of this paper is to design an adaptive PID controller using SGDM to control the DC/DC boost converter to achieve safe operation of the PEMFC system and to optimize the output power. The SGDM is used in order to set the PID adaptive gains according to the change of disturbances. Hence, the WOA algorithm is used to find the optimal adaptive rates, and is therefore injected into the SGDM technique.

This paper is divided into three sections. Section 2 describes the mathematical model of the PEM fuel cell. Section 3 is devoted to the control methodology. Section 4 presents the simulation results and the conclusion.

### 2. PEM Fuel Cell Modeling

As shown in Figure 1, a PEMFC is made up of two plates, two electrodes, and two thin layers of platinum-based catalysts separated by a membrane. When the fuel (hydrogen) is injected, it reacts electrochemically to create electricity [30]. The hydrogen and the oxygen are fed through channels in the plates. Hydrogen flows on one side of the membrane and the oxygen on the other. The catalyst splits the hydrogen molecule into protons and electrons; the protons can pass through the membrane, while the electrons cannot and must pass through an external circuit, creating useful electricity. On the oxygen side of the membrane, the protons and the electrons react with the oxygen in the presence of a second catalyst layer, generating water, heat, and electrical energy [31]. The reactions at the level of the PEMFC are given in Equations (1)–(3) [32]:

$$\text{Anode:} \quad 2H_2 \implies 4H^+ + 4e^- \tag{1}$$

$$\text{Cathode:} \quad 4H^+ + O_2 + 4e^- \implies 2H_2O \tag{2}$$

$$\text{Cell:} \quad 2H_2 + O_2 \implies 2H_2O + \text{E.E} \tag{3}$$

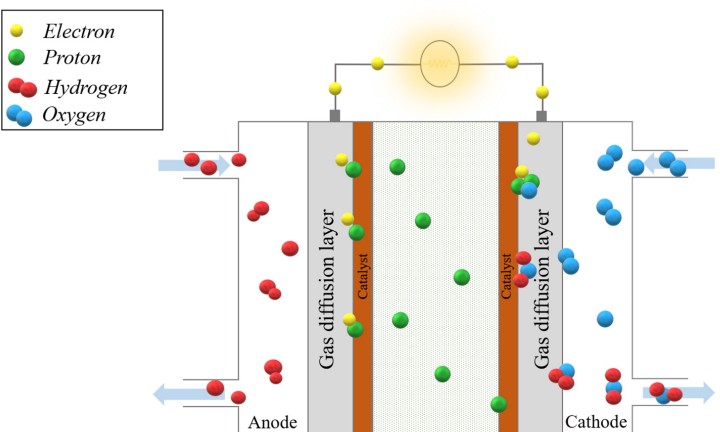

**Figure 1.** A cross-section of a PEMFC.

### 2.1. PEMFC Static Model

According to [32], the Nernst ($E_{Nern}$) equation describes the cell's electrochemical thermodynamic potential and gives the relationship between the open-circuit voltage of electrochemical cells under standard conditions and non-standard condition ($E_{Nop}$). The equation is as follows [32]:

$$E_{Nop} = 1.299 - 0.85 \cdot 10^{-3} \cdot (T - T_r) + 4.3085 \cdot 10^{-5} T \left[ ln(P_{H_2}) + \frac{1}{2} \cdot ln(P_{O_2}) \right] \qquad (4)$$

where $T$ is the cell operating temperature, $T_r$ represents the reference temperature in Kelvin, which is equal to 298.15 K at 25 °C, and $P_{O_2}$ and $P_{H_2}$ represent the inlet oxygen and hydrogen gas pressures, respectively [32].

The following expression can define the output voltage of a single PEMFC [33].

$$V_{Sfc} = E_{Nop} - E_{Act} - E_{Ohm} - E_{Con} \qquad (5)$$

where $E_{Act}$, $E_{Ohm}$, and $E_{Con}$ represent the polarization potentials or the voltage losses that are generated by the reversibility of the system.

The activation polarization $E_{Act}$ is due to the kinetics of the reactions taking place at the electrode/membrane reaction interface. This loss can be calculated using Equation (6) [33]:

$$E_{act} = \gamma_1 + \gamma_2 \cdot T + \gamma_3 \cdot T \cdot ln(C_{O_2}) + \gamma_4 \cdot T \cdot ln(I_{fc}) \qquad (6)$$

where the parameters $\gamma_1$, $\gamma_2$, $\gamma_3$, and $\gamma_4$ are the parametric coefficients for each PEMFC model; $C_{O_2}$ is the oxygen concentration in the catalysts (mol/cm$^3$).

The $E_{Ohm}$ polarization is caused by the electrical resistance of the different elements of the cell. This loss has two origins: the equivalent resistance of the membrane to proton conduction $R_{mem}$ and the contact resistance $R_{con}$ between the bipolar plates and the electrodes. The $E_{Ohm}$ voltage can be calculated using Equation (7) [33]:

$$E_{Ohm} = I_{fc} \cdot (R_{mem} + R_{con}) \qquad (7)$$

where

$$R_{mem} = \frac{\Gamma_{mem} \cdot l}{A} \qquad (8)$$

where $l$ is the membrane thickness (μm), $A$ is the cell active area (cm$^2$), and $\Gamma_{mem}$ is the specific resistance of the membrane, which is obtained by the following [34]:

$$\Gamma_{mem} = \frac{181.6[1 + 0.03(\frac{I_{fc}}{A}) + 0.062(\frac{T}{303})^2(\frac{I_{fc}}{A})^{2.5}]}{[\psi - 0.634 - 3(\frac{I_{fc}}{A})] \cdot \exp[4.18(T - 303)/T]} \qquad (9)$$

where $\psi$ is the water content in the membrane, assuming a minimum and maximum value of 0 and 24, respectively.

The concentration polarization $E_{Con}$ is caused by the variation in the concentration of reagents on the electrode. This loss can be calculated using Equation (10) [34]:

$$E_{con} = \delta \cdot ln \left( 1 - \frac{J}{J_{max}} \right) \qquad (10)$$

where $\delta$, $J$, and $J_{max}$ are the constant parameters, the current density, and the maximum current density, respectively.

A single PEMFC output voltage under standard conditions does not exceed 1.29 V. In order to produce the required amount of power, it is necessary to have cells in series, which finally forms a stack. Thereafter, the power generated by the PEMFC stack is given in Equation (11) [35]:

$$P_{stack} = V_{Sfc} \cdot I_{fc} \cdot N_{Cell} \qquad (11)$$

where $I_{fc}$ represents the single cell current and $N_{Cell}$ represents the number of stack layers. The PEMFC static model is represented in Figure 2.

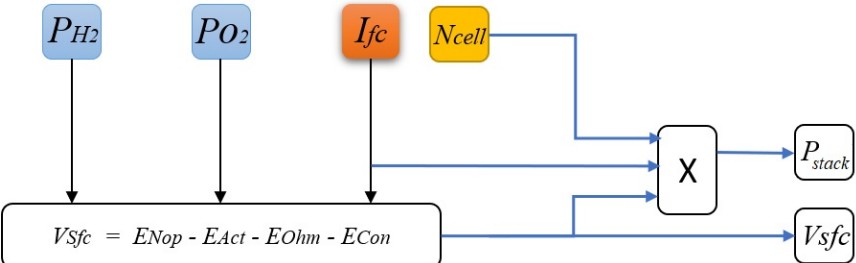

**Figure 2.** PEMFC static model.

The PEMFC parameters used in the simulation are given in Table 1:

**Table 1.** PEMFC parameters.

| Parameter | Value |
| --- | --- |
| $A$ | 162 cm$^2$ |
| $l$ | $175 \cdot 10^{-6}$ cm |
| $\psi$ | 23 |
| $\delta$ | 0.1 V |
| $R_{con}$ | 0.0003 |
| $J_{max}$ | 0.062 A·cm$^{-1}$ |
| $N_{Cell}$ | 10 |
| $\gamma_1$ | 0.9514 V |
| $\gamma_2$ | $-0.00312$ V/K |
| $\gamma_3$ | $-7.4 \cdot 10^{-5}$ V/K |
| $\gamma_4$ | $1.87 \cdot 10^{-4}$ V/K |

### 2.2. PEMFC Dynamic Model

In a PEMFC, the two electrodes are separated by a solid membrane that allows protons to pass and blocks the flow of electrons. The electrons flow from the anode through the external charge and are collected at the surface of the cathode, to which the hydrogen protons are attracted at the same time. Thus, two charged layers of opposite polarities are formed across the porous boundary between the cathode and the membrane. The layers are known as "double electrochemical layers" and can store electrical energy and behave like a super capacitor [36]. The PEMFC equivalent circuit showing this effect is presented in Figure 3.

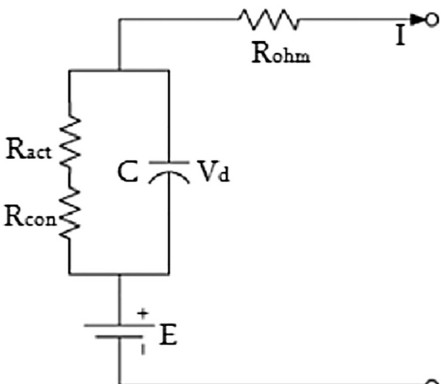

**Figure 3.** PEMFC equivalent circuit.

The electrodes of a PEMFC are porous. The capacitance is very large and can be of the order of several farads. $R_{act}$ and $R_{conc}$ are the activation equivalent resistance and the concentration equivalent resistance, respectively [37]. By using Kirchhoff's law, the dynamical equation of the model is represented by [38]:

$$\frac{dV_d}{dt} = \left( \frac{I}{C} - \frac{V_d}{\tau} \right) \tag{12}$$

where $V_d$ is the dynamical voltage across the capacitor, $C$ is the equivalent capacitor, and $\tau$ is the PEMFC time constant, which is given by the following equation [38]:

$$\tau = C \cdot (R_{act} + R_{conc}) = C \cdot \left( \frac{E_{Act} + E_{Con}}{I} \right) \tag{13}$$

Therefore, the PEMFC voltage is given by the equation below [38,39]:

$$V_{Sfc} = E_{Nern} - V_d - IR_{ohm} \tag{14}$$

where $R_{ohm,act,conc}$ represent the ohmic, activation, and concentration resistances, respectively.

Using Equations (12) and (13) and the Laplace transformations in Equation (14), the PEMFC voltage is given as follows [39]:

$$V_{Sfc} = E_{Nern} - \left( \frac{E_{Act} + E_{Con}}{sC \cdot (E_{Act} + E_{Con}) + 1} + R_{ohm} \right) \cdot I \tag{15}$$

According to [39], PEMFC inlet gas pressures are variable in different conditions. In order to calculate the dynamic partial pressures, each individual gas is considered separately and the ideal gas equation is applied for each one [40]. The partial gas pressures are given as follows [41]:

$$P_{H_2} = \frac{\frac{1}{K_{H_2}}}{(1 + \tau_{H_2})} \cdot (q_{H_2} - 2 \cdot I \cdot K_r) \tag{16}$$

$$P_{O_2} = \frac{\frac{1}{K_{O_2}}}{(1 + \tau_{O_2})} \cdot (q_{O_2} - 2 \cdot I \cdot K_r) \tag{17}$$

where

$$\begin{cases} \tau_{H_2} = \frac{V_{an}}{R \cdot T \cdot K_{H_2}} \\[2mm] \tau_{O_2} = \frac{V_{an}}{R \cdot T \cdot K_{O_2}} \end{cases} \tag{18}$$

By using the previous equations, the PEMFC dynamic model can be represented, as in Figure 4.

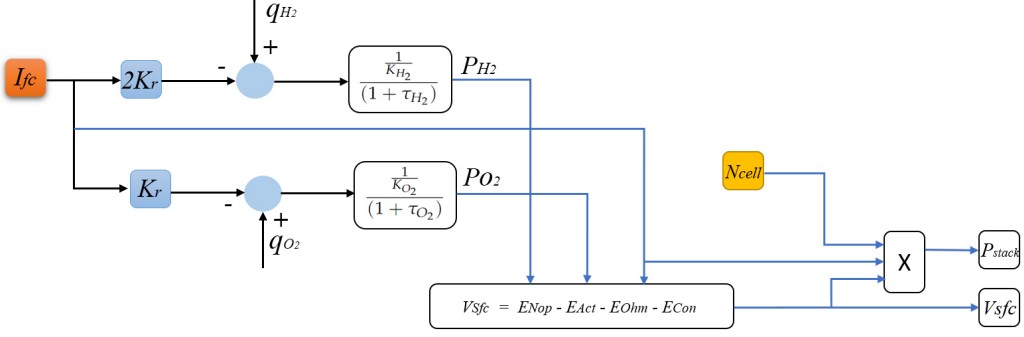

**Figure 4.** PEMFC dynamic model.

The meanings of the variables used in Equations (16)–(18) are listed in Table 2.

**Table 2.** PEMFC nomenclature.

| Variable | Meaning |
|---|---|
| $K_{H_2}$ | Hydrogen valve molar constant (kmol/atm·s) |
| $K_{O_2}$ | Oxygen valve molar constant (kmol/atm·s) |
| $\tau_{H_2}$ | Hydrogen time constant (s) |
| $\tau_{O_2}$ | Oxygen time constant (s) |
| $q_{H_2}$ | Molar flow rate of hydrogen (Kmol/s) |
| $q_{O_2}$ | Molar flow rate of oxygen (Kmol/s) |
| $K_r$ | Modeling constant (Kmol/s·A) |
| $R$ | Universal gas constant (1·atm/Kmol·K) |
| $V_{an}$ | Volume of the anode (cm$^3$) |

## 3. Control Methodology

In this section, an adaptive PID using SGDM and SGD is designed in order to stabilize the PEMFC and keep it operating at a reference current $I_{ref}$. The closed loop system consists of a PEMFC stack, a DC/DC boost converter, a *P&O* MPPT technique, a controller, and finally, a load, as shown in Figure 5.

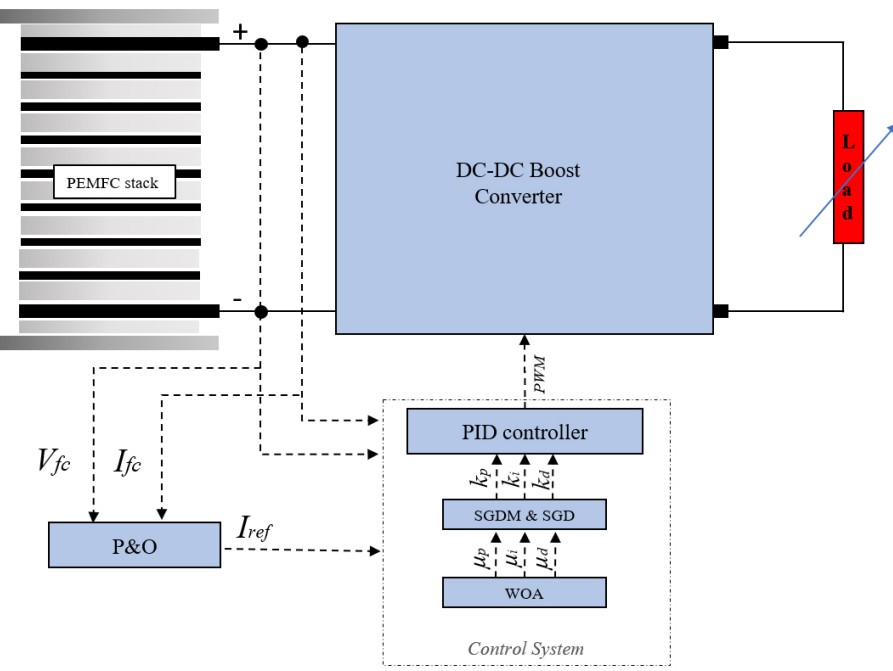

**Figure 5.** Closed loop system.

### 3.1. DC/DC Boost Converter

A step-up DC/DC converter is a simple electronic circuit that combines a switching element with a coil, a capacitor, and a diode, as shown in Figure 6. This converter works in order to obtain a DC output voltage higher than the input voltage. The important point to understand about the circuit is that the positions of the switching element (transistor), coil, and diode are different [42]. When the switch is turned *ON* and the current flows in, the coil stores energy; when the switch is turned *OFF*, the stored energy is released and the induced current flows in a direction that prevents the current change. The longer the switch is *ON*, the higher the output voltage is, and the longer the switch is *OFF*, the lower the output voltage is. The required output voltage can be obtained by controlling the *ON/OFF* time (duty cycle). The boost converter state space is given in Equation (19) [43]:

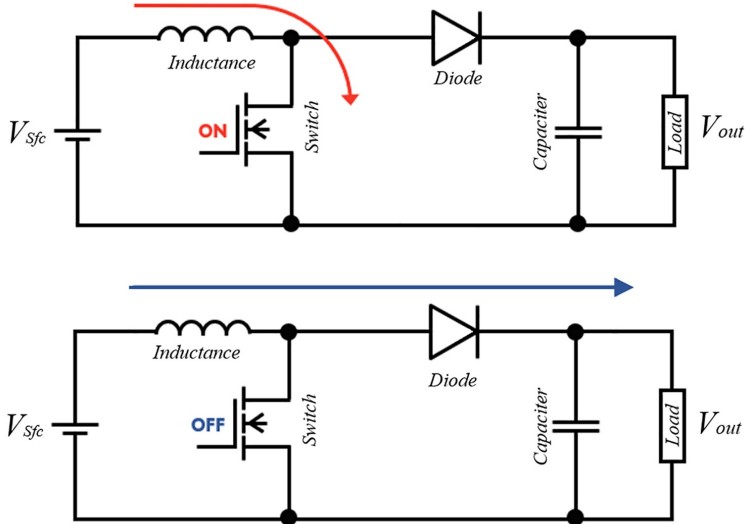

**Figure 6.** Step-up DC/DC converter diagram.

$$\begin{cases} \begin{bmatrix} \dot{x}_1 \\ \dot{x}_2 \end{bmatrix} = \begin{bmatrix} 0 & \frac{-(1-d)}{L} \\ \frac{(1-d)}{C} & -\frac{1}{RC} \end{bmatrix} \cdot \begin{bmatrix} x_1 \\ x_2 \end{bmatrix} + \begin{bmatrix} \frac{1}{L} \\ 0 \end{bmatrix} V_{Sfc} \\ \\ y = \begin{bmatrix} 0 & 1 \end{bmatrix} \cdot \begin{bmatrix} x_1 \\ x_2 \end{bmatrix} \end{cases} \tag{19}$$

where $x = [x_1, x_2]^T = [i_L, V_{out}]^T$.

The step-up DC/DC converter parameters used in the simulation are illustrated in Table 3.

**Table 3.** DC/DC boost converter parameters.

| Parameter | Value |
|---|---|
| Inductance (L) | $6.9 \cdot 10^{-2}$ H |
| Capacitor (C) | $150 \cdot 10^{-7}$ F |
| Max $f_{Sw}$ | 10 kHz |
| Max in voltage | 25 V |
| Max in current | 18 A |
| Max out voltage | 80 V |
| Max out current | 2 A |

### 3.2. Adaptive PID Using SGD

The adaptation optimum method used for the PID controller, known as SGD, consists of iteratively adjusting all of the proposed controller coefficients according to their calculated gradients in order to minimize the error function. The discreet PID general form is given as follows [44]:

$$u(k) = u(k-1) + K_p[e(k) - e(k-1)] + K_i e(k) + K_d[e(k) - 2e(k-1) + e(k-2)] \tag{20}$$

where $K_p$, $K_i$, and $K_d$ represent the proportional, integral, and derivative gains, respectively, and $k$ is the time instance.

The general mathematical formula for the SGD is given as follows [45]:

$$w(k+1) = w(k) - \mu \frac{\partial \mathcal{L}(k)}{\partial w(k)} \tag{21}$$

where $\mu \in (0,1)$ is the adaptive rate and $\mathcal{L}$ is the loss function, which is defined as follows [45]:

$$\mathcal{L}(k) = \frac{1}{2}e(k)^2 = \frac{1}{2}(r(k) - y(k))^2 = \frac{1}{2}(I_{ref}(k) - I_L(k))^2 \tag{22}$$

Therefore, the updated gains of the PID controller using SGD are given by the following equation:

$$K_{p,i,d}(k+1) = K_{p,i,d}(k) - \mu_{p,i,d}\frac{\partial \mathcal{L}(k)}{\partial K_{p,i,d}(k)} \tag{23}$$

where subscripts $p$, $i$, and $d$ indicate the values of $K_p$, $K_i$, and $K_d$, respectively. Equation (23) can be expressed by the following formula:

$$\Delta K_{p,i,d}(k) = -\mu_{p,i,d}\frac{\partial \mathcal{L}(k)}{\partial K_{p,i,d}(k)} = -\mu_{p,i,d}\frac{\partial \mathcal{L}(k)}{\partial I_L(k)}\frac{\partial I_L(k)}{\partial u(k)}\frac{\partial u(k)}{\partial K_{p,i,d}(k)} \tag{24}$$

Using Equations (20) and (22), the discrete partial derivative terms of Equation (24) can be expressed by the following formulas:

$$\begin{cases} \frac{\partial \mathcal{L}(k)}{\partial I_L(k)} = -e(k) = -(I_{ref}(k) - I_L(k)) \\ \frac{\partial I_L(k)}{\partial u(k)} = \frac{I_L(k) - I_L(k-1)}{u(k) - u(k-1)} \\ \frac{\partial u(k)}{\partial K_p(k)} = e(k) - e(k-1) \\ \frac{\partial u(k)}{\partial K_i(k)} = e(k) \\ \frac{\partial u(k)}{\partial K_d(k)} = e(k) - 2e(k-1) + e(k-2) \end{cases} \tag{25}$$

Therefore, by substituting formulas of Equation (25) into Equation (24), the adaptive gains can be expressed as follows:

$$\Delta K_p(k) = \mu_p \cdot e(k) \cdot \frac{I_L(k) - I_L(k-1)}{u(k) - u(k-1)} \cdot [e(k) - e(k-1)] \tag{26}$$

$$\Delta K_i(k) = \mu_i \cdot e(k) \cdot \frac{I_L(k) - I_L(k-1)}{u(k) - u(k-1)} \cdot e(k) \tag{27}$$

$$\Delta K_d(k) = \mu_d \cdot e(k) \cdot \frac{I_L(k) - I_L(k-1)}{u(k) - u(k-1)} \cdot [e(k) - 2e(k-1) + e(k-2)] \tag{28}$$

### 3.3. Adaptive PID Using SGDM

While SGD is a very popular optimization method, its learning process can sometimes be slow. The momentum method is designed to speed up learning (Figure 7), especially when dealing with gradients with high curvature, small but consistent gradients, or noisy gradients [46].

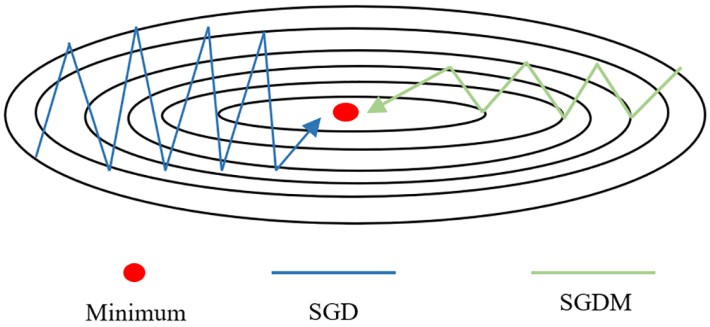

**Figure 7.** SGDM vs. SGD.

The momentum method continues to progress in the same path after accumulating the moving average of the preceding exponential decay of the gradient. The SGDM considers adding a momentum term to the basic Equation (21). Thus, an iteration form with momentum is written as follows [46,47]:

$$\begin{cases} v(k+1) = \beta v(k) - \mu \frac{\partial \mathcal{L}(k)}{\partial w(k)} \\ w(k+1) = w(k) + v(k+1) \end{cases} \tag{29}$$

Therefore, the updated gains of the PID controller using SGDM are given by the following equation:

$$\begin{cases} v(k+1) = \beta v(k) - \mu_{p,i,d} \frac{\partial \mathcal{L}(k)}{\partial K_{p,i,d}(k)} \\ K_{p,i,d}(k+1) = K_{p,i,d}(k) + v(k+1) \end{cases} \tag{30}$$

where $\beta$ is the attenuation coefficient, which is usually set to 0.9, and $v$ is the moving average of the gradients [48]. At each stage of the update, the algorithm adds the stochastic gradient to the old momentum value after dampening it by a factor $\beta$.

In order to calculate the update PID rules using SGDM, we substitute Equations (26)–(28) into Equation (30). Therefore, the adaptive gains can be expressed as follows:

$$\Delta K_p(k) = \beta v(k) + \mu_p \cdot e(k) \cdot \frac{I_L(k) - I_L(k-1)}{u(k) - u(k-1)} \cdot [e(k) - e(k-1)] \tag{31}$$

$$\Delta K_i(k) = \beta v(k) + \mu_i \cdot e(k) \cdot \frac{I_L(k) - I_L(k-1)}{u(k) - u(k-1)} \cdot e(k) \tag{32}$$

$$\Delta K_d(k) = \beta v(k) + \mu_d \cdot e(k) \cdot \frac{I_L(k) - I_L(k-1)}{u(k) - u(k-1)} \cdot [e(k) - 2e(k-1) + e(k-2)] \tag{33}$$

In order to enhance the SGDM and SGD adaptive rates $\mu_{p,i,d}$, a WOA is used, which is described in the next section.

### 3.4. Whale Optimization Algorithm

The WOA is an innovative heuristic optimization method that models the foraging activities of humpback whales. The location of each humpback whale is a "practical solution" according to the WOA algorithm [49]. In the realm of aquatic life, humpback whales are known for their unique hunting approach, which is known as the bubble-net predating strategy. An illustration of this whale hunting behavior is represented in Figure 8:

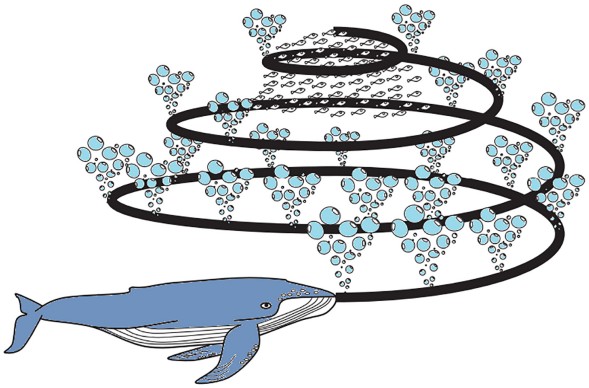

**Figure 8.** Humpback whale hunting behavior.

According to [49], the WOA hunting strategy takes place on three levels, which are as follows:

- Surrounding the prey;
- Bubble-net attacking method;



- Searching for prey.

When hunting, humpback whales encircle their prey. In order to provide an explanation for this pattern of behavior, Mirjalili suggested the following mathematical model [49,50]:

$$\overrightarrow{D} = |\overrightarrow{C}\overrightarrow{X^*}(k) - \overrightarrow{X}(k)| \tag{34}$$

$$\overrightarrow{X}(k+1) = |\overrightarrow{X^*}(k) - \overrightarrow{A}.\overrightarrow{D}| \tag{35}$$

where $k$ denotes the current iteration number, $\overrightarrow{A}$ and $\overrightarrow{C}$ express the coefficient vectors, $\overrightarrow{X^*}$ represents the best whale position vector found so far, and $\overrightarrow{X}(k)$ denotes the position vector of the current whale. The vectors $\overrightarrow{A}$ and $\overrightarrow{C}$ can be obtained by the following formula [50]:

$$\overrightarrow{A} = 2.\overrightarrow{a}.\overrightarrow{r} - \overrightarrow{a} \tag{36}$$

$$\overrightarrow{C} = 2.\overrightarrow{r} \tag{37}$$

During the iterations, the value of the vector $\overrightarrow{a}$ decreases from 2 to 0 in a linear manner and the value of the vector $\overrightarrow{r}$ is a random value between 0 and 1.

### 3.4.1. Bubble-Net Attacking Method

In humpback whale hunting habit, a whale approaches its prey using a spiraling motion. The mathematical model of the hunting activity is given as follows:

$$\overrightarrow{X}(k+1) = \overrightarrow{D} \cdot e^{bl} \cdot cos(2 \cdot \pi \cdot l) + \overrightarrow{X^*}(k) \tag{38}$$

where $\overrightarrow{D} = |\overrightarrow{X^*}(k) - \overrightarrow{X(k)}|$ denotes the distance between the $i$th whale and the prey, $b$ is a constant that specifies the shape of the logarithmic spiral, and $l$ is a random value that falls within the range $[-1, 1]$ [49,50].

Remarkably, humpback whales swim in a spiraling circle around their prey. To represent this simultaneous behavior, assuming a 50% chance of selecting between a diminishing encirclement mechanism and a spiral model to optimize the whale's location, the mathematical model is as follows [49]:

$$\overrightarrow{X}(k+1) = \begin{cases} \overrightarrow{X^*}(k) - \overrightarrow{A} \cdot \overrightarrow{D} & if \quad \beta < 0.5 \\ \overrightarrow{D} \cdot e^{bl} \cdot cos(2 \cdot \pi \cdot l) + \overrightarrow{X^*}(k) & if \quad \beta > 0.5 \end{cases} \tag{39}$$

where $\beta$ is a random number between $[0, 1]$ [49].

### 3.4.2. Search for Prey

Humpback whales undertake random searches depending on each other's locations, thus using random values larger than 1 or less than $-1$ to drive search agents away from reference whales. In contrast to the search for prey behavior, here, the position of the search agent is updated based on a randomly selected search agent, rather than the best search agent so far. This mechanism and $|\overrightarrow{A}| > 1$ exploration is emphasized and allows the WOA algorithm to perform a global search. The mathematical model is as follows [49]:

$$\overrightarrow{D} = |\overrightarrow{C}\overrightarrow{X_{rand}}(k) - \overrightarrow{X}| \tag{40}$$

$$\overrightarrow{X}(k+1) = \overrightarrow{X_{rand}} - \overrightarrow{A} \cdot \overrightarrow{D} \tag{41}$$

The WOA algorithm initially generates a set of solutions at random; then, in each iteration, the search agents update their locations in accordance with either the randomly chosen search agent or the best solution thus far. While $\overrightarrow{a}$ is reduced with the number of iterations, it changes from exploration to utilization gradually. When the optimal solution is selected, the search agent position is updated according to $p$. The WOA can switch between

helical and circular motion. Finally, the WOA algorithm is terminated by satisfying the termination criterion [49,51].

In this paper, the WOA is used in order to enhance the SGDM and SGD adaptive rates $\mu_{p,i,d}$ in the offline mode. The obtained adaptive rates are injected into both techniques. The WOA pseudo-code is shown in Figure 9 [52].

```
Initialize the whales population Xi=(1, 2, 3 ......, n)
Calculate the fitness function for each search agent
X*= the best search agent
While (k <maximum number of iteration)
    for each search agent
    Update a, A, C, l and p
      if1 (p<0,5)
        if2 (|A|<0,5)
          Update the position of the current search agent by the Eq (35)
        else if2 (|A| ⩾1)
          Select a random search agent (Xrand)
          Update the position of the current search agent by the Eq (41)
        end if2
      else if1 (p⩾0,5)
          Update the position of the current search by the Eq (38)
      end if1
    end for
    Check if any search agent goes beyond the search space and amend it
    Calculate the fitness of each search agent
    Update X* if there is better solution
    k=k+1
end while
Reurn X*
```

**Figure 9.** WOA pseudo-code.

In this simulation, the population size equals 40 search agents, and the number of iterations equals 100. The implementation of WOA is intended to minimize the fitness function ITAE [53], while the adaptive gains $\mu_{p,i,d}$, are taken as decision variables. The variation ranges of the decision variables used in the simulation are given in Table 4.

**Table 4.** WOA upper and lower bounds.

| Algorithm | Range | $\mu_p$ | $\mu_i$ | $\mu_d$ |
|-----------|-------|---------|---------|---------|
| WOA | Min | 0 | 0 | 0 |
| | Max | 1 | 1 | 1 |

## 4. Simulation Results

The main aim of this work is to design an adaptive PID using SGDM applied to a DC/DC step-up converter in order to keep the PEMFC power system working at a reference current $I_{ref}$. This simulation was done under fixed hydrogen and oxygen gas pressures equal to 2 bar. Furthermore, a variety of temperatures and loads (Figure 10) were applied in order to validate the control strategies robustness. The acquired parameters for the implemented controllers used in this study are given in Table 5.

**Table 5.** Controllers obtained parameters.

| Controller | $K_p$ | $K_i$ | $K_d$ | $\mu_p$ | $\mu_i$ | $\mu_d$ | $\beta$ |
|------------|-------|-------|-------|---------|---------|---------|---------|
| PIDSGDM | - | - | - | 0.0013 | 0.1678 | 0.0025 | 0.9 |
| PIDSGD | - | - | - | 0.0013 | 0.1678 | 0.0025 | - |
| PID-ZN | 0.03 | 10.6 | 0.003 | - | - | - | - |

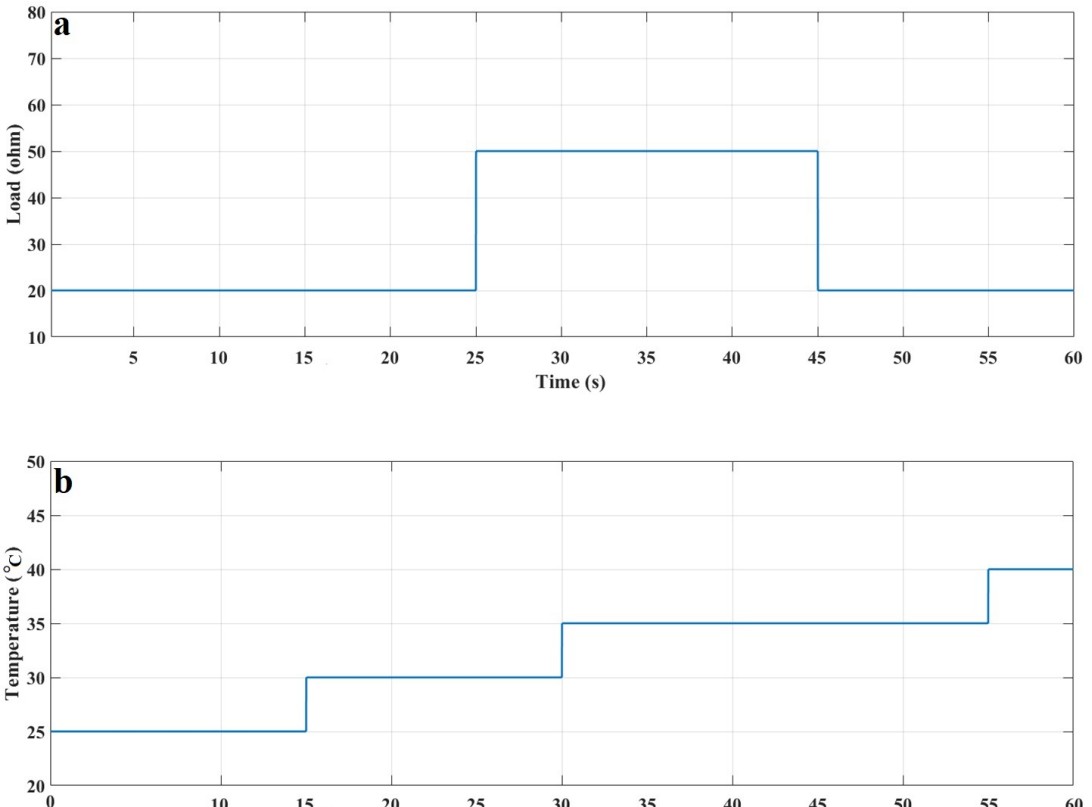

**Figure 10.** (**a**) Resistance; (**b**) temperature.

Figure 11 exhibits the behavior of the stack output current under the PIDSGDM, PIDSGD, and PID-ZN when the load and change in temperature are applied. In Figure 11, it is seen that the PIDSGDM has faster convergence than the other approaches. On the other hand, high performance in tracking the reference current is achieved by the PIDSGDM and PIDSGD. Going forward to $t = 15$ s, $t = 30$ s, and $t = 55$ s, the PIDSGDM and PIDSGD have almost the same overshoot, which is approximately equal to 0 A. Hence, during the same periods $t = 15$ s, $t = 30$ s, and $t = 55$ s, the PID-ZN shows an overshoot equal to 0.09 A, 0.0231 A, and 0.069 A, respectively. On the other hand, at $t = 25$ s, the PIDSGD and PID-ZN show an undershoot of 0.097 A and 1.175 A, respectively, while there is an undershoot of 0.065 A with the PIDSGDM; this results in a difference 33% and 94% higher compared to the PIDSGD and PID-ZN, respectively. Moving to the period $t = 45$ s, the PIDSGD and PID-ZN show an overshoot of 0.1 A and 1.297 A, respectively, while there is an overshoot of 0.083 A with the PIDSGDM; this results in a difference 17% and 93% higher compared to the PIDSGD and PID-ZN, respectively.

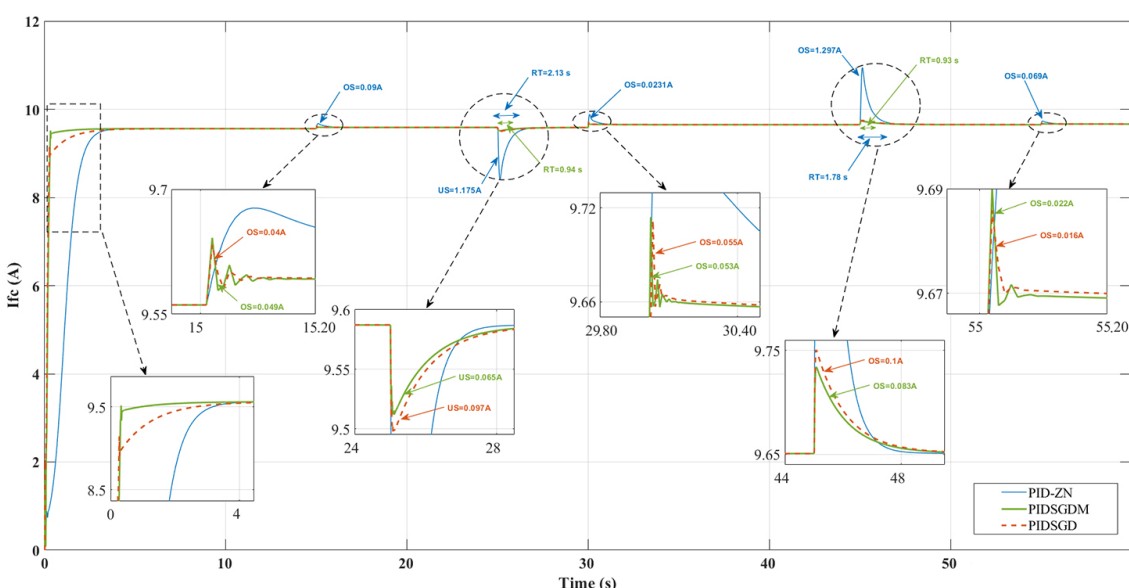

**Figure 11.** PEMFC stack output current.

Figure 12 exhibits the behavior of the stack output voltage under the PIDSGDM, PIDSGD, and PID-ZN. According to this figure, it is noticeable again that the PIDSGDM has faster convergence than the other approaches. On the other hand, at $t = 15$ s, $t = 30$ s, and $t = 55$ s, both the PIDSGDM and PIDSGD controllers have almost the same undershoot, which is approximately equal to 0 V. At the same periods $t = 15$ s, $t = 30$ s, and $t = 55$ s, the PID-ZN shows an undershoot equal to 0.025 V, 0.072 V, and 0.018 V, respectively. Going forward to $t = 25$ s, the PIDSGD and PID-ZN show an overshoot of 0.028 V and 0.371 V, respectively, there is an overshoot of 0.021 V with the PIDSGDM; this results in a difference 25% and 94% higher compared to the PIDSGD and PID-ZN, respectively. Moving on to the period $t = 45$ s, the PIDSGD and PID-ZN show an undershoot of 0.031 V and 0.111 V, respectively, while there is an undershoot of 0.022 V with the PIDSGDM; this results in a difference 29% and 80% higher compared to the PIDSGD and PID-ZN, respectively.

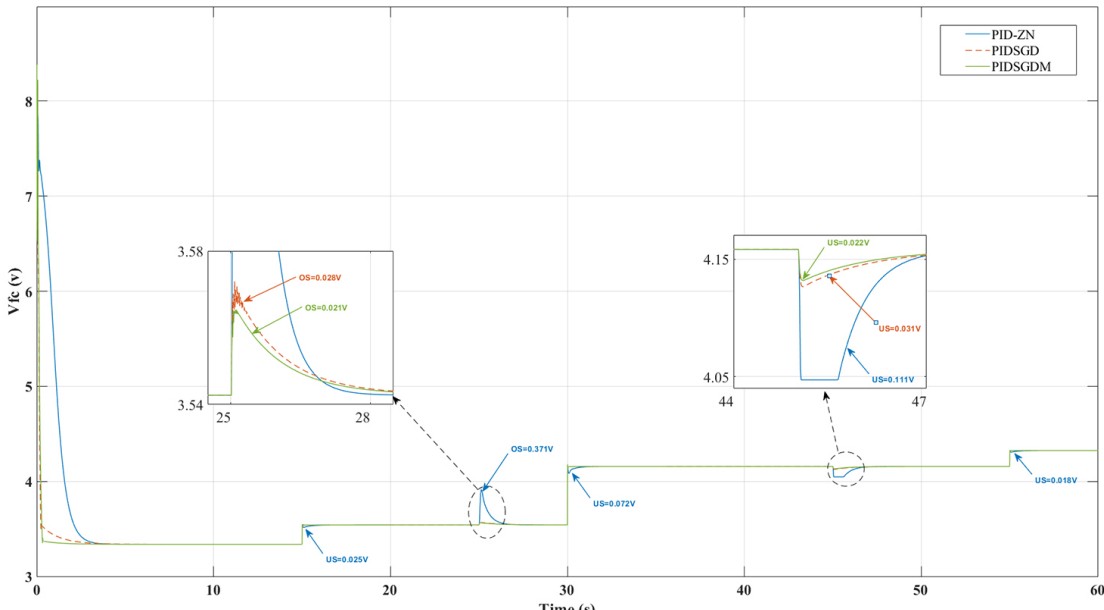

**Figure 12.** PEMFC stack output voltage.

Figure 13 exhibits the behavior of the stack output power under the PIDSGDM, PIDSGD, and PID-ZN. An analysis of the overshoot at $t = 15$ s, $t = 30$ s, and $t = 55$ s indicated that the PIDSGDM and PIDSGD controllers have almost the same overshoot, which is approximately equal to 0 W. At the same periods $t = 15$ s, $t = 30$ s, and $t = 55$ s, the PID-ZN shows an overshoot equal to 0.06 W, 0.24 W, and 0.12 W, respectively. Going forward to $t = 25$ s, the PIDSGD and PID-ZN show an undershoot of 0.07 W and 1.02 W respectively, while there is an undershoot of 0.05 W with the PIDSGDM; this results in a difference 29% and 95% higher compared to the PIDSGD and PID-ZN, respectively. Moving on to the period $t = 45$ s, the PIDSGD and PID-ZN show an overshoot of 0.11 W and 4.18 W, respectively, while there is an overshoot of 0.09 W for the PIDSGDM; this results in a difference 18% and 98% higher compared to the PIDSGD and PID-ZN, respectively.

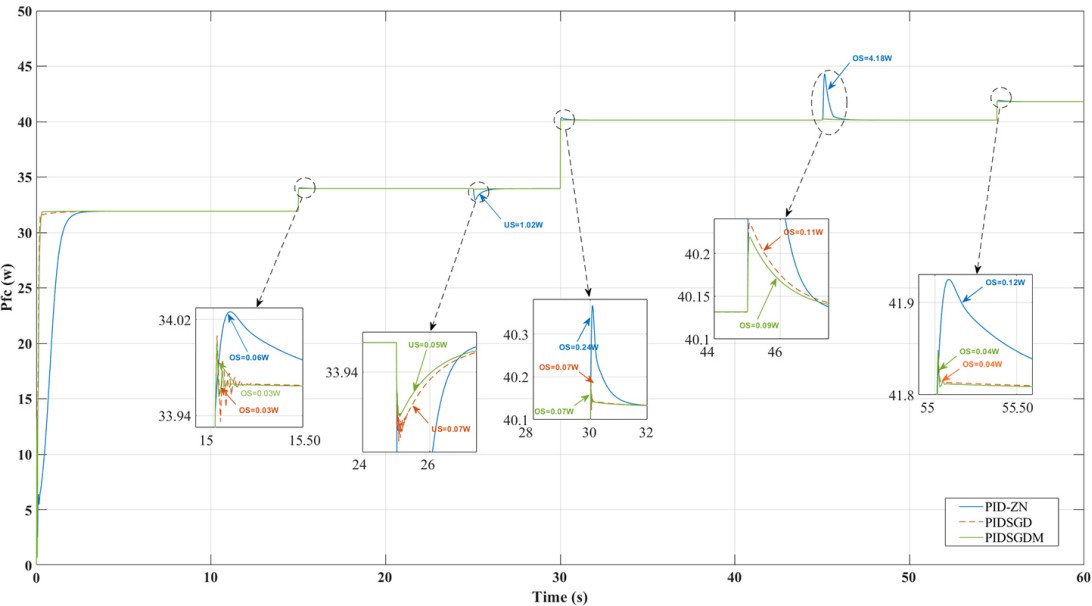

**Figure 13.** PEMFC stack output power.

According to Figures 11–13, the robustness of the PIDSGDM, PIDSGD, and PID-ZN controllers is apparent at $t = 25$ s and $t = 45$ s. Thus, despite the extreme load variation, the PIDSGDM shows high robustness against this perturbation. Table 6 shows the performance of different control techniques comparing the response time and overshoot and undershoot reductions.

**Table 6.** Performance comparison between previous works.

| Work | Response Time (s) | Overshoot % | Undershoot % | Controller |
|---|---|---|---|---|
| Proposed | 0.94 | 93 | 94 | PIDSGDM and PID-ZN |
| [20] | 0.92 | 42.15 | 46 | QC-HOSM and SMC |
| [43] | 0.45 | 1.32 | 0.44 | IFTSMC and PI |
| [23] | 1.2 | 34 | 8.47 | HOSMC-TA and SMC |
| [7] | 1 | 33 | 33 | STA and SMC |
| [29] | 0.51 | 13.7 | 17.6 | MPC and PI |

In addition to these results, the PIDSGDM robustness analysis is used regarding the variation of circuit parameters. As the components are commonly influenced by uncertainties and perturbation, this type of analysis is required in order to evaluate the control performances. The effects of circuit parameter variation (capacitance and inductance) on the fuel cell current are represented in Figure 14. According to this figure, the PIDSGDM is robust for both cases of parameter variation, which ranged from −50% to +50%. In more

detail, the increase of capacitance leads to a change in the system dynamics and slower system response. In the opposite case, when the capacitance decreases, system response is faster and there is a residual ripple increase.

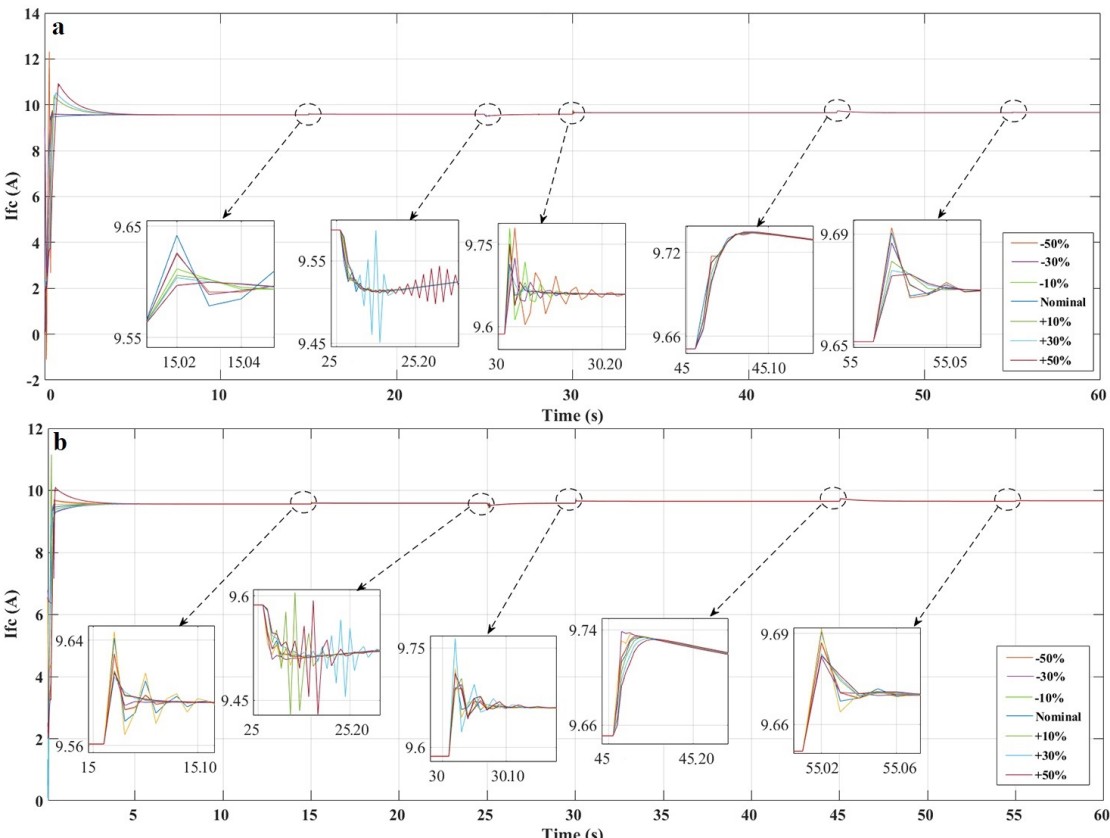

**Figure 14.** (**a**) PEMFC current control robustness analysis: effect of L variation; (**b**) PEMFC current control robustness analysis: effect of C variation.

Figure 15 exhibits the demeanor of the boost converter output signals under PIDS-GDM, PIDSGD, and PID-ZN. According to this figure, smooth and gradual movements to the desired value are achieved using the PIDSGDM, PIDSGD, and PID-ZN. Furthermore, PIDSGDM, and PIDSGD have fast convergence, which is clearly presented in this figure. On the deep side, according to the current and power results, it is notable that the three controllers have approximately the same performance in terms of overshoot and undershoot. On the other hand, the PIDSGDM and PIDSGD can successfully bear the voltage overshoot, unlike the PID-ZN controller. However, these overshoots and undershoots can theoretically be considerably reduced by using an extra capacitor on the input side of the converter. Nevertheless, adding extra capacitance may lead to a change in the system dynamics, slower system response, and steady-state errors.

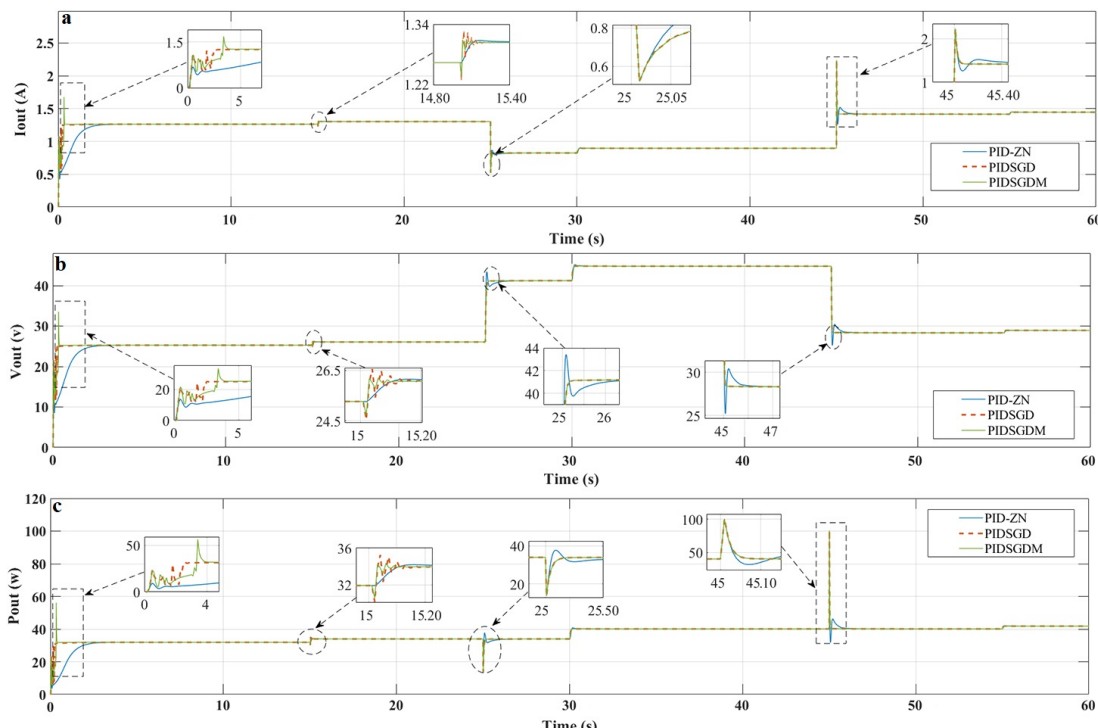

**Figure 15.** (**a**) Step-up converter output current; (**b**) step-up converter output voltage; (**c**) step-up converter output power.

The obtained adaptation gains $K_p$, $K_i$, and $K_d$ for the PIDSGDM and PIDSGD are shown in Figure 16.

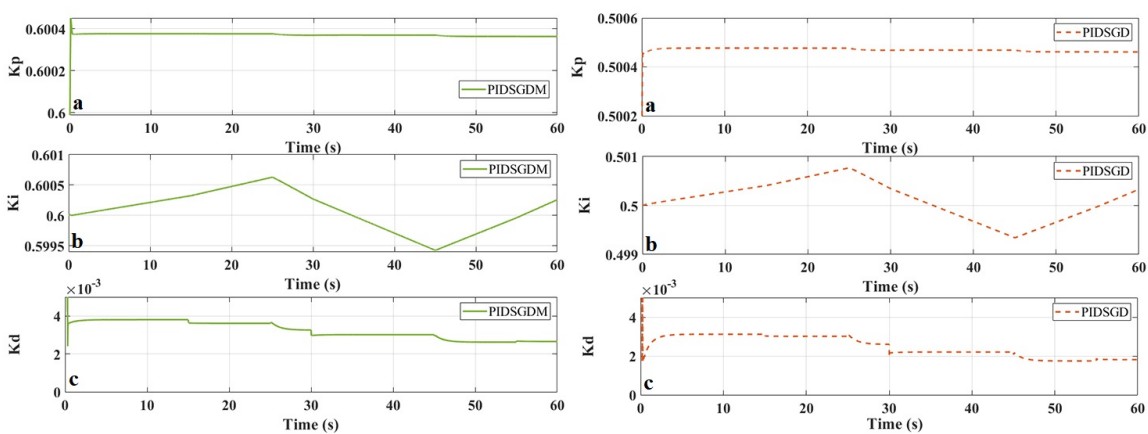

**Figure 16.** (**a**) Adaptive gain $K_p$; (**b**) adaptive gain $K_i$; (**c**) adaptive gain $K_d$.

According to these results, the proposed PIDSGDM dynamic performance, especially fast-tracking time, avoids the gas starvation phenomena that occur when gas transmission lags behind the speed of current change, and there is no apparent performance effect on the external characteristics such as the output voltage and output power of the PEMFC. This ability is more significant in the case of stepped variation, which is considered the hardest case.

During the simulation, the control performance was further validated using a Simulink profiler [15,54]. The profiler collects performance data while simulating and creates a report based on this data, known as a simulation profile, which provides the amount of Simulink time spent performing each function to simulate the model. The profile assists in identifying the model components that need the most time to simulate, and

thus, where to concentrate model optimization efforts. According to the Simulink profiler, the computational complexity terms of the control strategies are summarized in Table 7. As remarked, the complexity terms of the control techniques are approximately similar, ranging between 0.88%–1.22%, which is a very low complexity term compared to the whole model.

**Table 7.** Computational complexity summary.

| Algorithm | Time | Calls | Time/Call | Self Time |
|---|---|---|---|---|
| PIDSGDM | 0.058–1.22% | 600127 | 0.0000000966 | 0.057–2.19% |
| PIDSGD | 0.051–1.07% | 570155 | 0.0000000897 | 0.048–1.84% |
| PID-ZN | 0.020–0.88% | 18005 | 0.0000011 | 0.0020–0.33% |

## 5. Conclusions

In this paper, an adaptive PID using SGDM was proposed that is effective and has a low complexity of implementation. This is applied to a simple DC/DC boost converter in order to achieve safe operation of a PEMFC system and to optimize the output power quality. The proposed controller was compared with the PIDSGD and PID-ZN techniques. Based on the results, an overshoot reduction up to 98% and an undershoot reduction up to 94% were achieved. The simulation comparison demonstrates that the PIDSGDM can bear disturbances and offers low response time, high robustness, and overshoot and undershoot, which leads to a minimization of the power losses. Consequently, as is demonstrated in the results, the proposed controller provides safe operation and an optimal solution for a PEMFC system affected by external disturbances.

**Author Contributions:** Conceptualization, M.Y.S.; methodology, M.Y.S.; software, M.Y.S.; validation, M.Y.S., O.B. and A.B.; formal analysis, M.Y.S. and A.B.; investigation, M.Y.S., O.B. and A.B.; resources, O.B.; data curation, M.Y.S. and A.B.; writing—original draft preparation, M.Y.S.; writing—review and editing, M.Y.S., O.B. and A.B.; visualization, M.Y.S., O.B. and A.B.; supervision, O.B.; project administration, O.B.; funding acquisition, O.B. All authors have read and agreed to the published version of the manuscript.

**Funding:** The authors wish to express their gratitude to the Basque Government through the project EKOHEGAZ (ELKARTEK KK-2021/00092), to the Diputación Foral de Álava (DFA) through the project CONAVANTER, and to the UPV/EHU through the project GIU20/063 for supporting this work.

**Acknowledgments:** The authors wish to express their gratitude to the Basque Government through the project EKOHEGAZ (ELKARTEK KK-2021/00092), to the Diputación Foral de Álava (DFA) through the project CONAVANTER, and to the UPV/EHU through the project GIU20/063 for supporting this work.

**Conflicts of Interest:** The authors declare no conflict of interest.

## Abbreviations

The following abbreviations are used in this manuscript:

| | |
|---|---|
| SGD | Stochastic gradient descent |
| SGDM | Stochastic gradient descent with momentum |
| WOA | Whale optimization algorithm |
| ZN | Ziegler Nichols |
| PEMFCs | Proton exchange membrane fuel cells |
| SMC | Sliding mode control |
| PI | Proportional–integral |

| MPC | Model predictive control |
|---|---|
| FOPID | Fractional-order proportional-integral-derivative |
| MPPT | Maximum power point technique |
| PSO | Particle swarm optimization |
| PID | Proportional integral derivative |
| P&O | Perturb and observe |
| PWM | Pulse-width modulation |
| FL | Fuzzy logic |
| GWO | Grey wolf optimizer |
| EGWO | Extended grey wolf optimizer |

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
