# Peer review of "A Novel Adaptive PID Controller Design for a PEM Fuel Cell Using Stochastic Gradient Descent with Momentum Enhanced by Whale Optimizer"

_electronics, doi:10.3390/electronics11162610_

Round 1

Reviewer 1 Report

The subject matter is certainly interesting and the work is decently presented.

For my part, there are just a few suggestions on the introductory part and some comments on the simulations presented.

I suggest that the authors divide the introduction section into several subsections, which would make it more readable.

The following subdivision for example:

1.1 Motivations, where the authors explain the problem they want to solve, is because it has practical technical importance, mentioning the limitations of the current state of the art.

1.2 State of the arts and/or Related works, in which the authors describe and directly compare existing solutions highlighting where they intend to make a contribution.

In this subsection, I would mention that there are already some works in the literature where adaptive control based on gradient descent, applied to power electronic systems, is presented. For example (but not limited to only that work):

https://www.mdpi.com/1996-1073/13/10/2512

1.3 Contributions, in which the authors make an explicit list of the contributions that this work introduces with respect to the state of the art.

Regarding the simulations presented.

Being a work done purely in simulation, I strongly recommend that the authors increase the analysis of the control algorithm.

The factual simulations show "only" the comparison between existing solutions and the proposed algorithm. Which in simulation is clearly advantageous.

The problem is that all simulations are carried out under nominal conditions (according to the values in Table 2).

Since there is no section devoted to real experiments, I think it is necessary (if not mandatory) to present an analysis of parametric robustness and variation of inputs.

Showing what happens if the values of the circuit components vary within an acceptable percentage and what happens if the input voltage has fluctuations.

The assumption that the input voltage is impressed (i.e., always constant, as if supplied by an always-charged battery) is undoubtedly widely accepted and used in modelling, however, it is not entirely realistic.

I recommend that the authors take their cue from the following work, presenting a robustness analysis for a power electronic circuit.

https://www.mdpi.com/2079-9292/11/1/112

I also recommend taking a cue from the following work:

https://ieeexplore.ieee.org/document/9805740

to develop a discussion of the effect of numerical methods on your proposed algorithm, presenting simulation results under the same operating condition. 

I also recommend an analysis of computational complexity using the Simulink Profiler, as reported in one of the suggested papers, with some comments on possible embedded platforms suitable for real-world implementation.

I hope the authors find my comments helpful in improving the presentation of the work, which is of good quality anyway.

Author Response

Response to the reviewer comments

First of all, the author would express their sincere gratitude to the Editors and the Reviewers who gave us many constructive comments and valuable suggestions in order to improve this paper. The authors have revised the paper according to the reviewers’ comments and the changes made in the paper have been written in blue color. While the mistakes have been depicted in red color. The responses to the reviewer comments can be found below their respective comments.

Reviewer 1

The subject matter is certainly interesting and the work is decently presented.

For my part, there are just a few suggestions on the introductory part and some comments on the simulations presented.

Point 1:  I suggest that the authors divide the introduction section into several subsections, which would make it more readable.

The following subdivision for example:

1.1 Motivations, where the authors explain the problem they want to solve, is because it has practical technical importance, mentioning the limitations of the current state of the art.

1.2 State of the arts and/or Related works, in which the authors describe and directly compare existing solutions highlighting where they intend to make a contribution.

In this subsection, I would mention that there are already some works in the literature where adaptive control based on gradient descent, applied to power electronic systems, is presented. For example (but not limited to only that work):

https://www.mdpi.com/1996-1073/13/10/2512

1.3 Contributions, in which the authors make an explicit list of the contributions that this work introduces with respect to the state of the art.

Response 1: The authors have taken into consideration the reviewer comments about the introduction subdivisions. Hence, the authors have added subdivisions to the revised manuscript.  

Point 2:   Regarding the simulations presented.

Being a work done purely in simulation, I strongly recommend that the authors increase the analysis of the control algorithm.

Response 2:  Thank you for your comment. The authors have increased the analysis of the algorithm in section simulation results (line 358-361) and (364-368) for the revised manuscript.  

Point 3:   The factual simulations show "only" the comparison between existing solutions and the proposed algorithm. Which in simulation is clearly advantageous.

The problem is that all simulations are carried out under nominal conditions (according to the values in Table 2).

Response 3:  Thank you for the comment. All controllers’ techniques were done at the same nominal conditions. And applied the same perturbation as clearly mentioned in section simulation results (a variety of temperatures and loads (Figure 10), in order to test the robustness of the controllers. Moreover, it should be noted that this theoretical study has been done based on the technical data of a commercial Heliocenris fuel cell system in order to work with a model of a commercial Fuel cell system.

Moreover, it should be noted that the proposed controller do not use the system model parameters for the controller tuning. Then, in real applications the controller will be tuned with the real system not with the system model so there are not modelling errors.

Point 4:  Since there is no section devoted to real experiments, I think it is necessary (if not mandatory) to present an analysis of parametric robustness and variation of inputs.

Response 4: The authors declare fixed hydrogen and oxygen gas pressures; those parameters are taken according to the real PEMFC Heliocenris. Please see attached articles. Where the main aim of this work is to validate the controller under load and temperature variations. 

https://www.mdpi.com/1996-1073/13/17/4317

https://www.mdpi.com/2071-1050/13/4/2360

https://www.sciencedirect.com/science/article/pii/S0306261921008606

https://www.sciencedirect.com/science/article/abs/pii/S0360319916310916

Moreover, taking into account the reviewer comments a robustness analysis are added (line 346-350) to the new version of manuscript.

Point 5: Showing what happens if the values of the circuit components vary within an acceptable percentage and what happens if the input voltage has fluctuations.

Response 5: Thank you for the comment. The input voltage generated by the PEMFC has been affected by a change of load from 20 Ohm to 100 Ohm at t= 25s and from 100 Ohm to 20 Ohm at t=45s. According to these variations, the input voltage has some fluctuations as shown in Figure 14 (zooms are added to this figure). Besides, the fluctuations can be varied by the variation of DC/DC boost converter capacitance as mention in line (360-361) for the revised manuscript.

Point 6: The assumption that the input voltage is impressed (i.e., always constant, as if supplied by an always-charged battery) is undoubtedly widely accepted and used in modelling, however, it is not entirely realistic.

I recommend that the authors take their cue from the following work, presenting a robustness analysis for a power electronic circuit.

https://www.mdpi.com/2079-9292/11/1/112

I also recommend taking a cue from the following work:

https://ieeexplore.ieee.org/document/9805740

Response 6: Thank you for the comment. In order to test the controller’s robustness a variety of perturbations are applied such as temperatures and load changes (as presented in Figure 10). Hence, those perturbation are mentioned in the abstract section, simulation results section, besides a robustness analyses are added (line 346-350) to the new version of manuscript.

to develop a discussion of the effect of numerical methods on your proposed algorithm, presenting simulation results under the same operating condition. 

I also recommend an analysis of computational complexity using the Simulink Profiler, as reported in one of the suggested papers, with some comments on possible embedded platforms suitable for real-world implementation.

Response 7:

The authors have taken into consideration the reviewer comments about the computational complexity. Hence, the authors have added computational complexity  (369-379) to the revised manuscript. 

I hope the authors find my comments helpful in improving the presentation of the work, which is of good quality anyway.

Response 8:

The authors thanks to the reviewer his/her valuable comments.

Reviewer 2 Report

The authors present the article entitled “A novel adaptive PID controller design for a PEM fuel cell using stochastic gradient descent with momentum enhanced by whale optimizer”

This paper represents an adaptive PID controller using the stochastic gradient descent with momentum (SGDM) for the proton exchange membrane fuel cell (PEMFC). 

The article presents the following concerns:

The following misspelling should be checked:

  1. The Abstract section must be restructured. This section should give a pertinent overview of the work in a single paragraph. Please check the guide for authors.

  2. Avoid using apostrophes in the manuscript.

  3. Introduction section: It is presented several works related to the , what is the novelty of the proposal work?

  4. Self-tuning neural network pid with dynamic response control

  5. A PID-type fuzzy logic controller-based approach for motion control applications

  6. Plese introduce the PID controller by considering the following fresh references, this in line 38: Self-tuning neural network pid with dynamic response control; A PID-type fuzzy logic controller-based approach for motion control applications.

  7. The MPPT could be referenced in line 57 with the paper Artificial neural networks in mppt algorithms for optimization of photovoltaic power systems: A review.

  8. PEM devices coulbe threatened before line 27, for example, considenring the reference Electrochemical hydrogen production using separated-gas cells for soybean oil hydrogenation.

  9. line 88-93: The objective of the article must be improved by higlighting the novelty of the work.

  10. Figure 7: Please use a) and b) labels for each plot instead top and bottom. Also, in this figure, the temperature unit must be °C (capital letter)

  11. Figures 11 and 12: Please describe each subplot in detail. Also, use labels a) b) and c) for each plots.

  12. Include a table which compares the findings of the work vs the already reported in the stat of the art.

  13. Include quantitative results in the last section.

  14. line 41: “has still...” should be rewritten as “still has…”

  15. line 58: “proportional integral derivative (PID) controller in compare with perturb and observe…” should be rewritten as “a proportional integral derivative (PID) controller compared with perturbing and observing…”

  16. page 4: “it’s…” Contractions such as it’s may be too informal for this writing style. Consider replacing it with an uncontracted form: “It is”

  17. Page 10: “....to gradually change from exploration to utilization…” should be rewritten as “it changes from exploration to utilization gradually.”

  18. page 11: “In this simulation, the population size is equal to 40 search agents, and the number of iterations is equal to 100” your sentence may be unclear or hard to follow. Consider rephrasing  by “In this simulation, the population size equals 40 search agents, and the number of iterations equals 100”

  19. page 15: “low undershoot, and low…” should be rewritten as “undershoot, and…”

Author Response

Response to the reviewer comments

First of all, the author would express their sincere gratitude to the Editors and the Reviewers who gave us many constructive comments and valuable suggestions in order to improve this paper. The authors have revised the paper according to the reviewers’ comments and the changes made in the paper have been written in blue color. While the mistakes have been depicted in red color. The responses to the reviewer comments can be found below their respective comments.

Reviewer 2

The authors present the article entitled “A novel adaptive PID controller design for a PEM fuel cell using stochastic gradient descent with momentum enhanced by whale optimizer”

This paper represents an adaptive PID controller using the stochastic gradient descent with momentum (SGDM) for the proton exchange membrane fuel cell (PEMFC). 

The article presents the following concerns

The following misspelling should be checked:

  1. The Abstract section must be restructured. This section should give a pertinent overview of the work in a single paragraph. Please check the guide for authors

Response: The authors have taken into consideration the reviewer comment about the abstract section. Hence, the authors have reform the abstract (revised manuscript).

  1. Avoid using apostrophes in the manuscript.

Response: The authors have taken into consideration the reviewer comment about the apostrophes. Hence, the apostrophes have been removed from the revised manuscript.

  1. Introduction section: It is presented several works related to the, what is the novelty of the proposal work?

Self-tuning neural network pid with dynamic response control. A PID-type fuzzy logic controller-based approach for motion control applications

Response: Thank you for the comment. The contribution of the paper has been clearly presented in the introduction of the revised manuscript (please see lines 100-104 revised manuscript). We think that the paper has an important contribution since it proposes a novel adaptive PID based on SGD with momentum technique enhanced by whale algorithm, aiming to improve the performance of the fuel cell system in terms of robustness, and convergence. A comparison study of two optimized algorithms (PIDSGD and PID-ZN) has been performed to validate the proposed algorithm performance. Moreover, it should be noted that this theoretical study has been done based on the technical data of a commercial Heliocentris fuel cell system in order to facilitate the real implementation of this control scheme as a future work.

  1. Please introduce the PID controller by considering the following fresh references, this in line 38: Self-tuning neural network pid with dynamic response control; A PID-type fuzzy logic controller-based approach for motion control applications.

Response: The authors have taken into consideration the reviewer comment about the PID controller references. Hence, the references are updated in the new version of the manuscript.

  1. The MPPT could be referenced in line 57 with the paper Artificial neural networks in mppt algorithms for optimization of photovoltaic power systems: A review.

Response: The authors have updated the proposed references in the new version of the manuscript according to the reviewer comments.

  1. PEM devices could be threatened before line 27, for example, considering the reference Electrochemical hydrogen production using separated-gas cells for soybean oil hydrogenation.

Response: The authors have updated the proposed references in the new version of the manuscript according to the reviewer comments.

  1. line 88-93: The objective of the article must be improved by higlighting the novelty of the work.

Response: The authors have taken into consideration the reviewer comment about the article objective. Hence, the objective is improved (line 100-104) in the new version of the manuscript.

  1. Figure 7: Please use a) and b) labels for each plot instead top and bottom. Also, in this figure, the temperature unit must be °C (capital letter)

Response: The authors have taken into consideration the reviewer comment about Figure 7. Hence, the labels and the temperature unit are added in (Figure 10) of the new version of the manuscript.

  1. Figures 11 and 12: Please describe each subplot in detail. Also, use labels a) b) and c) for each plots.

Response: The authors have taken into consideration the reviewer comment about Figure 11 and Figure 12. Hence, the subplots and labels are added in (Figure 14 and Figure 15) of the new version of the manuscript.

  1. Include a table which compares the findings of the work vs the already reported in the stat of the art.

Response: The authors have taken into consideration the reviewer comment about the comparative table. Hence, a comparative table (Table 6) is added in the last section of the new version of manuscript.

  1. Include quantitative results in the last section.

Response: The authors have taken into consideration the reviewer comment about the quantitative results. Hence, a quantitative result is added in the last section of the new version of manuscript.

  1. line 41: “has still...” should be rewritten as “still has…”

Response:  Thank you for the comment. The statement “has still” is revised in the line 48 of the revised manuscript as follows: “still has”.

  1. line 58: “proportional integral derivative (PID) controller in compare with perturb and observe…” should be rewritten as “a proportional integral derivative (PID) controller compared with perturbing and observing…”

Response: Thank you for the comment. The statement “proportional integral derivative (PID) controller in compare with perturb and observe…”” is revised in the line 65 of the revised manuscript as follows: “a proportional integral derivative (PID) controller compared with perturbing and observing”.

  1. page 4: “it’s…” Contractions such as it’s may be too informal for this writing style. Consider replacing it with an uncontracted form: “It is”

Response: Thank you for the comment. The statement “it’s…” is revised in page 5 of the revised manuscript as follows: “it is…”.

  1. Page 10: “....to gradually change from exploration to utilization…” should be rewritten as “it changes from exploration to utilization gradually.”

Response: Thank you for the comment. The statement “..to gradually change from exploration to utilization…”” is revised in page 12 of the revised manuscript as follows: “it changes from exploration to utilization gradually”.

  1. page 11: “In this simulation, the population size is equal to 40 search agents, and the number of iterations is equal to 100” your sentence may be unclear or hard to follow. Consider rephrasing by “In this simulation, the population size equals 40 search agents, and the number of iterations equals 100”

Response: Thank you for the comment. The statement “In this simulation, the population size is equal to 40 search agents, and the number of iterations is equal to 100”” In this simulation, the population size equals 40 search agents, and the number of iterations equals 100”.

  1. page 15: “low undershoot, and low…” should be rewritten as “undershoot, and…”

Response: The authors have taken into consideration the reviewer comment. Hence, the authors have reform the conclusion (revised manuscript).

Round 2

Reviewer 1 Report

The article was already quite well presented, so I don't feel like giving another major for the latest shortcomings.

However, I am not satisfied with the response regarding the robustness analysis in terms of variation in circuit parameters. I recommended replicating the robustness analysis reported in this article

https://www.mdpi.com/2079-9292/11/1/112

Being a simulation paper, I do not think it is a great effort for the authors to generate results with different values of some of the parameters and superimpose the curves to show the reader that the algorithm is robust to a full range of variability. 

This point I think should be included and adds value to the presentation of the article.

I note that the choice of initial conditions, which is an important point in adaptive algorithms, is never discussed. They, rightly, state that tuning is not based on the mathematical model because it is done "online." However, I imagine that the mathematical model helps in the choice of initial conditions.  

I also see that in the end there is no discussion of the variation in the numerical method chosen to describe the dynamics of fitting the control parameters. I think this is an important enough point to integrate with the Simulink Profiler results.

These last 2 points are definitely more challenging and also I realize they are more "technical quibbles," more suitable for a formal discussion of robustness properties. In this paper, there is a pragmatic technical discussion but not too formal, because that is not the real focus of the authors' work.

However, I leave it to the discretion of the authors to incorporate these latter comments, as I consider the work suitable for publication in MDPI.

Author Response

The authors would like to thank the Editorial Board of Electronics Journal, Respective Editor Handling and Reviewers for their valuable suggestions, comments and efforts to review this paper. The authors have revised faithfully the manuscript according to the editor and reviewers’ comments. The authors believe that the manuscript has been greatly improved and hope it has reached your expectation.

Once again, the authors acknowledge your comments very much, which are valuable in improving the quality of our manuscript.

The corrections in the manuscript have been depicted in the blue-colored text. While the mistakes in the manuscript have been depicted in the red-colored text.

Reviewer 1:

Point: The article was already quite well presented, so I don't feel like giving another major for the latest shortcomings.

However, I am not satisfied with the response regarding the robustness analysis in terms of variation in circuit parameters. I recommended replicating the robustness analysis reported in this article

https://www.mdpi.com/2079-9292/11/1/112

Being a simulation paper, I do not think it is a great effort for the authors to generate results with different values of some of the parameters and superimpose the curves to show the reader that the algorithm is robust to a full range of variability. 

This point I think should be included and adds value to the presentation of the article.

I note that the choice of initial conditions, which is an important point in adaptive algorithms, is never discussed. They, rightly, state that tuning is not based on the mathematical model because it is done "online." However, I imagine that the mathematical model helps in the choice of initial conditions.  

I also see that in the end there is no discussion of the variation in the numerical method chosen to describe the dynamics of fitting the control parameters. I think this is an important enough point to integrate with the Simulink Profiler results.

These last 2 points are definitely more challenging and also I realize they are more "technical quibbles," more suitable for a formal discussion of robustness properties. In this paper, there is a pragmatic technical discussion but not too formal, because that is not the real focus of the authors' work.

However, I leave it to the discretion of the authors to incorporate these latter comments, as I consider the work suitable for publication in MDPI.

Response: Thank you for your comments. The authors have taken into consideration the reviewer's comments about the robustness analysis in terms of variation in circuit parameters. Hence, the authors have increased the analysis to the revised manuscript (Figure 14) and (Line 351 to 359).

Reviewer 2 Report

The manuscript is ready for publishing.

Author Response

Reviewer 2:

Point: The manuscript is ready for publishing.

Response: The authors would like to thank the Reviewer for his valuable suggestions, comments, and efforts to review this paper.
